# Greenhouse Gas Emissions and Energy Consumption of Coastal Ecosystem Enhancement Programme through Sustainable Artificial Reefs in Galicia

**DOI:** 10.3390/ijerph18041909

**Published:** 2021-02-16

**Authors:** Luis Carral, Juan José Cartelle Barros, Humberto Carro Fidalgo, Carolina Camba Fabal, Alicia Munín Doce

**Affiliations:** 1Escola Politécnica Superior, Universidade da Coruña, 15403 Ferrol, Spain; l.carral@udc.es (L.C.); carolina.camba@udc.es (C.C.F.); a.munin@udc.es (A.M.D.); 2Escola Técnica Superior de Enxeñaría de Camiños, Canais e Portos, Universidade da Coruña, 15071 A Coruña, Spain; humberto.fidalgo@udc.es

**Keywords:** artificial reefs, ecosystem enhancement, global warming potential, cumulative energy demand, supply chain

## Abstract

The principle of sustainability should condition a project in which artificial reefs are being installed to protect biodiversity as well as enhance costal ecosystems. In particular, this principle should be taken into account in the logistical processes related to manufacture and transport. This study assesses the global warming potential (*GWP*) and cumulative energy demand (*CED*) of developing a coastal ecosystem enhancement programme in the estuary region of Galicia, north-western Spain. The focus is on the processes involved in creating green artificial reefs (GARs): manufacture, transport and installation. The starting point is the supply chain for the green artificial reef (*GAR*) units; greenhouse gas emissions (*GHG*) and energy needs for each phase are analysed. Various scenarios are considered to determine which options are indeed available when it comes to establishing the supply chain. Different types of energy supplies, different options for the location of production centres, as well as different means of transport were studied. Results reveal the critical phases for selecting how the GAR units must be produced, transported by road and sea and then installed in their permanent location.

## 1. Introduction

### 1.1. Coastal Ecosystem Enhancement Programme in Galicia

The coastal regions of Galicia (NW Iberian Peninsula) stand out for their geographical features and for the economic resources they provide. Indeed, geography experts use the term “ría” in English because of the unique way in which the finger-like Galician estuaries are formed with the contrasting heights of the land and sea. In this singular environment, the ecosystem boasts a high level of natural resources: marine species make up the first link of the food chain and are crucial to a thriving local economy [1].

Nonetheless, Galician estuaries also need protection. On the one hand, the fishing and seafood gathering industries, which have elevated levels of production and excellent economic prospects, can have a negative effect on the environment. The impact of other human activity can also be felt, including building and water sports, among others [2]. With this situation, it is necessary to take measures that foster the richness and biodiversity of these areas [3].

One way to protect biodiversity and marine species is to install artificial reefs (AR). They behave in a similar way as their natural counterparts by offering a refuge for different species to reproduce. These are called production modules. At the same time, ARs in the form of protection modules provide a shield; fishing methods severely harmful to biodiversity are thus impeded [4]. For these and other reasons, the PROARR (Proyecto de Arrecife Artificial Reciclado or Recycled Artificial Reef Project in translation) project was launched with the aim of enhancing Galician coastal ecosystems by developing and manufacturing both production and protection green artificial reef (GAR) units. A GAR is an artificial reef (AR) in which some of the conventional materials are replaced by waste [5]. The reader can find in Table 1 the most relevant information of the GAR units proposed by the authors [5]. They were adapted to the geomorphology of the Galician estuaries [2,3,6]. The modules were also optimised hydrodynamically through computational fluid dynamics (CFD) [7]. 

A GAR (green artificial reef) programme is needed to enhance coastal ecosystems in Galicia. For such a purpose, groups of GAR units (GAR groups) are necessary for the entire estuary region. The reader should bear in mind that, at the time of producing a GAR group, a careful planning is a key factor for reducing the potential impacts on the environment (Figure 1). In fact, if production, transportation and installation processes of the GAR groups are not carried out in accordance with sustainability criteria, the potential positive impacts could be outweighed by negative ones. In this regard, Stenico de Campos et al. [8] argue that sustainability must be the lynchpin of all industrial and commercial processes, including manufacturing, maintenance, commercialisation, purchasing, sales, and logistics. However, not all of these headings have the same degree of importance, since they do not present the same potential economic, environmental and social impacts. 

Carral et al. [5] developed a multicriteria decision making model for assessing the sustainability of different GARs for the Galician coast. The authors considered the use of mussel shell residues and eucalyptus fibres as substitutes for conventional materials (cement, sand and steel frames). Economic, social and environmental indicators were included. Nevertheless, the authors did not analyse the potential environmental impacts derived from the manufacturing, transportation and installation processes. It is important to note that all these life cycle stages could be responsible for a large amount of carbon dioxide emissions and considerable energy consumption [9,10]. Consequently, from an environmental point of view, sustainability efforts should focus on manufacturing and logistics, the main gap to be overcome in this study. It is important to remark that great differences are not expected at the time of analysing the environmental impacts of ARs and GARs derived from the manufacturing, transportation and installations processes. Consequently, the terms AR and GAR are equivalent in this paper. Furthermore, the type of artificial reef analysed in this study is the production one. Protection reefs are expected to obtain similar results.

### 1.2. Objective

The main objective of this study is to assess both the global warming potential (*GWP*) and cumulative energy demand (*CED*) indicators associated with a programme to enhance the coastal ecosystems of the Galician estuaries. This programme is based on the use of green artificial reef (GAR) groups. Manufacturing, transport and installation of the units are within the scope of this study (Figure 1). Different types of energy supplies, different options for the location of production centres, as well as different means of transport will be considered. This will be of great help in determining the solution that releases the least amount of greenhouse gases (*GHG*), as well as the solution that requires the least amount of energy. To the best of the authors’ knowledge, this is the first time that the production and distribution life cycle stages of GARs are analysed from an environmental point of view. The results of this work together with the ones presented in previous studies [2,5] will serve to select the best option for enhancing Galician estuaries from the point of view of sustainability.

## 2. Materials and Methods

### 2.1. Environmental Impacts: General Information

As indicated in Section 1.2, global warming potential (*GWP*) and cumulative energy demand (*CED*) are the two environmental indicators used in this study. The reader should bear in mind that the European Union (EU) prioritises the reduction of both energy consumption and emissions [11]. On the one hand, the *GWP* indicator is an index for estimating the extent to which an activity has an impact on global warming through the atmospheric emission of greenhouse gases (*GHG*) [12] and the impact they have on climate change. This indicator is measured in kg of CO_2-eq_. 

On the other hand, *CED* is useful to measure the primary energy used throughout the entire supply chain for the reef units. In other words, this indicator calculates the energy that is used during a specific life cycle stage of a product [13]. It first appeared as a method at the beginning of the 1970s, after the first petroleum crisis [14,15]. *CED* is measured in megajoules (MJ).

Consequently, this study’s scope covers the moment in which the components for the GAR units arrive at the manufacturing plant to the final phase in which GAR units are transported by sea to their permanent locations fixed to the seabed (Figure 1). The process of obtaining the components used in concrete and their transport to the production plant do not fall within the scope of this analysis. It is important to remark that *GWP* and *CED* factors used in this study are a proposal from the authors based on the existing literature. All the results of this work are referred to a GAR unit (or to a certain number of GAR units). In other words, the GAR unit (or multiple of GAR units, in particular the number of units needed for restoring Galician estuaries) can be considered as the functional unit (FU) used here. The mass of each GAR unit is 7.5 tonnes (average mass). Consequently, it can also be said that the functional unit is a quantity of mass. According to Boschiero et al. [16], in a considerable number of life cycle analysis (LCA) studies the FU is based on a quantity of mass, since this makes it possible to easily compare results with those from other studies. 

It is assumed that all components will enter and exit the manufacturing plant. Thus, the energy consumption and *GHG* emissions are assigned to the total quantities of manufactured product, which is the only outcome of the system. The manufacturing and distribution system examined in this study is divided into the phases outlined in Figure 2.

### 2.2. Manufacturing, Transport and Installation Processes: General Information

The manufacturing of GAR units entails several phases within the supply chain (Figure 2). This can be done in two ways. On the one hand, they can be produced right in the port. On the other, the manufacturing can also be done in a centralised way [17]. Production plants are required to have a total available surface area large enough for manufacturing the units, as well as a space to reinforce and store them. Moreover, a curing process of 28 days is necessary after the manufacturing. Then, lifting devices will also be needed to load them [17].

Many of the areas in which the ecosystem enhancement will take place have estuaries with small fishing ports whose pier surface area is too small for manufacturing and storing the GAR units. Such piers can only be used for specific, one-off tasks.

If production is centralised (in one or several manufacturing plants), road transport is required to carry the units to the port. To minimise the distance travelled, Gayo Romero [17] recommends that the plant should be as close as possible to the port. Transport, in this study, can be carried out in two different ways. One is by articulated trucks with a two-axle tractor, three-axle trailer and a maximum permissible load capacity of 38 tonnes (t). The other option is a rigid truck with three axles and a maximum permissible load of 26 t [18]. This second vehicle works well along tracks and/or entrances to small ports unsuitable for articulated trucks.

In terms of maritime transport, the points of departure will be determined with the goal of minimising the time needed [3]. The lifting equipment available at the port will not be considered as the vessel that is deployed has its own gear for loading and unloading [17]. The following two systems of maritime transport are considered in this study: i) mini bulk carrier and, ii) special service work boat. As its name suggests, a mini bulk carrier is a small-sized carrier (81 m long) with cargo holds enhanced by its relatively high draft (5.45 m). The second type of vessel has a self-propulsion system, an auxiliary crane and on-deck storage. As its length is only 24 m and its draft is 2.3 m, it is ideal for the manoeuvres involved in port activity and the task of anchoring the reef units to the seabed. 

Small vessels are generally provided with lifting equipment suitable for loading and unloading at ports and piers unequipped with cranes. In the case of GAR units, a 10 t safe working load (SWL) is needed for the crane to carry out port activities and take part in fastening the reef at sea. While the work boat has on-deck storage, the mini bulk carrier uses a cargo hold with a sliding hatch. The different cargo systems affect the number of GAR units that can be stowed. However, the vessel cranes have similar capacities for hoisting and lowering. The reader can find in Table 2 more information related to the two types of vessels considered in this study.

For the installation process (positioning and anchoring), the vessel has to change its position, anchoring at different points. The mini bulk carrier is larger than the special service work boat, consequently, its manoeuvrability is expected to be hindered during the task of fixing the unit to the seabed. The system for lowering the GAR units is by means of a cable. Great precision is required during this stage. Consequently, it must be supervised by divers in constant communication with the vessel to guide the entire process.

### 2.3. Galician Estuaries and Related Information

López-Davalillo [19] classifies the Galician estuaries according to their geographical location and orography in Northern, Central and Southern Estuaries. All the Galician estuaries (Figure 3) are considered in this study.

It is also possible to make a more detailed classification by zones: (i) Northern zone including Ribadeo, Foz, Vivero, Barquero, Ortigueira, and Cedeira estuaries; (ii) Costa Ártabra formed by Ferrol, Ares, Betanzos, and La Coruña estuaries; (iii) Costa da Morte (or Costa de la Muerte in Spanish) with Corme y Lage, Camariñas, Corcubión, Muros, and Noya estuaries; (iv) Rías Bajas including Arousa, Pontevedra, Aldán, and Vigo estuaries. The Northern zone belongs to the Northern Estuaries, while Costa Ártabra and Costa da Morte are part of the Central Estuaries. Finally, Rías Bajas is the only zone of the Southern Estuaries.

Subindex *i* is used to distinguish among the four different zones. Subindex *j* is used for a similar purpose but, this time, for distinguishing among the estuaries of a specific zone. Each estuary needs a specific number of GAR groups. Each GAR group consists of 60 GAR units. Consequently, subindex *k* is used for differentiating among all GAR groups of a specific estuary. For instance, Ribadeo will need 180 GAR units, that is, three GAR groups. Therefore, for Ribadeo, *k* will vary between 1 and 3. Letter *A* is used for identifying a GAR group. For practical purposes, *A* always presents three subscripts (*i*, *j* and *k*: *A_i,j,k_*) with the objective of indicating the zone, the estuary as well as the number of the group, respectively. For example, Ribadeo is the first estuary (*j* = 1) of the Northern zone (*i* = 1) and it has three GAR groups: *A_1,1,1_*, *A_1,1,2_* and *A_1,1,3_*. As each GAR group has 60 GAR units, the following notation is used for identifying a specific GAR unit: *A_i,j,k,l_*. Subindex *l* always varies from 1 to 60. By way of example, *A_1,1,1,1_* is one of the 60 GAR units belonging to the first GAR group (*k* = 1) to be installed in Ribadeo (*j* = 1), first estuary of the Northern zone (*i* = 1).

Each estuary has an associated harbour *H*, that is, the point of departure. Consequently, all harbours present two subscripts *i* and *j* for identifying the zone and the estuary, respectively. In other words, *H_1,1_* is the harbour of the first zone (Northern zone, *i* = 1) for the first estuary (Ribadeo, *j* = 1). It is important to note that, on a practical level, two different estuaries can have the same harbour, as is the case for Ferrol, Ares, Betanzos, and La Coruña estuaries in Table 3 (*H_2,1_* = *H_2,2_* = *H_2,3_* = *H_2,4_* = Lorbé harbour), among others. All the GAR groups (*A_i,j,k_*) and their corresponding GAR units (*A_i,j,k,l_*) of a specific estuary have the same point of departure (*H_i,j_*). Nevertheless, each GAR group (*A_i,j,k_*) (and their corresponding GAR units) belonging to a specific estuary has its own point of arrival. Consequently, the points of arrive will be designated as *PA_i,j,k_*. It will be assumed that the point of arrival will be the same for all the GAR units belonging to a specific GAR group *k*.

The reader can find in Table 3 the most relevant information about the Galician estuaries for this study.

The enhancement of each region entails installing several GAR groups according to surface area and the peculiarities of each case. A total number of 4080 GAR units are needed for the Galician estuaries (1080 for the Northern zone, 840 for Costa Ártabra, 1020 for Costa da Morte, and 1140 for Rías Bajas). All GAR units (*A_i,j,k,l_*) are transported to the corresponding port (*H_i,j_*) from the manufacturing plant. Two scenarios are considered for the manufacturing process. In the first case, each zone *i* has its own manufacturing plant. Consequently, the manufacturing plant will be designated as *MP_i_*. In the second one, the same manufacturing centre is used for all zones. It will be noted as *CMP* (centralised manufacturing plant).

Taking into account all the above, each GAR unit (*A_i,j,k,l_*) belonging to a specific GAR group (*A_i,j,k_*) must be transported from the manufacturing plant (*MP_i_* or *CMP*) to the harbour (*H_i,j_*) and, after that, to the point of arrival in the sea (*PA_i,j,k_*). The first is a road transport in which the distance only depends on the estuary *j* and the zone *i*. In the case of the second scenario, subindex *i* is only relevant for the harbour, since the location of the manufacturing plant is always the same for the four zones. The distance between the manufacturing plant and the harbour will be noted as *Ld_i,j_*. The second is a maritime transport. In this case, the point of arrival (*PA_i,j,k_*) also changes according to subindex *k*, as explained before. In other words, the distance between a specific harbour (*H_i,j_*) and the point of arrival of a specific GAR group will depend on subscripts *i*, *j* and *k*. Consequently, it will be noted as *Md_i,j,k_*. For estimating *Ld_i,j_* it is first necessary to determine the exact location of the manufacturing plants for both scenarios. This will be explained later in Section 2.4.

The reader can find in Figure 4 an example of scheme concerning the location of some hypothetical manufacturing plants (*MP_i_* and *CMP*), a harbour (*H_i,j_*) and the point of arrival (*PA_i,j,k_*) for a specific GAR group (*A_i,j,k_*).

### 2.4. Determining the Location of the Manufacturing Plant

As previously indicated in Section 2.3, two scenarios are considered (*MP_i_* and *CMP*). For both cases, a static location method is employed. This method is based on the use of the centre of gravity to determine the coordinates [20]. It pinpoints the most economic location for the plant by taking into account both volume and transport costs [21].

In most cases, the location determined by this method is not ideal, but it is a good starting point. Thus, once the theoretical location has been obtained by using the corresponding equations [20], only one more step needs to be taken. In other words, nearby real industrial areas must be located, since, in most cases, there is no point in building the plant on the theoretical coordinates. In fact, it must be built on the industrial estate closest to the theoretical coordinates.

### 2.5. Determining GWP and CED

In the particular case of the manufacturing process, it is necessary to consider the energy inputs (*EI_t_*) across the different phases (*t*). They are included in Table 4. These figures consider the amount of energy directly used in manufacturing, that is, the electricity consumed by the use of the machinery. Other energy inputs associated with the extraction and transport of raw materials, building the manufacturing plant or the disposal of waste, among others, are out of the scope of this study.

It is now necessary to determine the emission factor for *GWP* (*GWP_electricity_*) as well as the consumption or use factor for *CED* (*CED_electricity_*) associated with the production of 1 kWh of electricity. The Spanish electricity grid mix is used for such a purpose [22], and the corresponding values are included in Table 5. It is important to note that these figures are proposed by the authors on perusing the literature [23,24,25,26,27,28]. They are representative of Spain, the country in which this study takes place.

Therefore, *GWP* for manufacturing 1 GAR unit is estimated from Equation (1) and it is measured in kg CO_2-eq._/FU.
(1)$$GWP_{Manufacturing}=\overset{9}{\underset{t=1}{\sum }}EI_{t}\cdot GWP_{electricity}$$

Similarly, *CED* for manufacturing 1 GAR unit is calculated from Equation (2). The units of measurement are MJ/FU.
(2)$$CED_{Manufacturing}=\overset{9}{\underset{t=1}{\sum }}EI_{t}\cdot CED_{electricity}$$

It is also possible to estimate the total value of both indicators (measured in kg of CO_2-eq._ and MJ, respectively) derived from the complete enhancement programme in the estuary region of Galicia (4080 GAR units):(3)$$Total-GWP_{Manufacturing}=4080\cdot GWP_{Manufacturing}$$
(4)$$Total-CED_{Manufacturing}=4080\cdot CED_{Manufacturing}$$

On the other hand, a second scenario is considered in which only wind, hydroelectric, solar photovoltaic, and solar thermal alternatives participate in the Spanish electric grid mix. This is an ideal scenario that will be used to determine the extent to which results could be improved by using the most common renewable sources. It is important to note that this scenario may not be realistic. The corresponding emission factors (*GWP_electricity−ideal_*, *CED_electricity−ideal_*) associated with the production of 1 kWh of electricity are included in Table 6.

*GWP* and *CED* for manufacturing 1 GAR under this alternative scenario can be estimated by using Equations (1) and (2), replacing *GWP_electricity_* and *CED_electricity_* by *GWP_electricity−ideal_* and *CED_electricity−ideal_*, respectively. After that, Equations (3) and (4) can also be applied.

On the other hand, transport and installation phases can also present a considerable impact in terms of both *GWP* and *CED*. In this sense, this study only takes into account direct energy outputs and emissions. In other words, other related activities such as building or recycling the vessels, among others, are out of the scope of this work. Road and maritime transport will be analysed separately. It is a question of converting the distance covered and the time spent on loading (including stowage of cargo), unloading and installation operations (positioning and anchoring the GAR units) into an amount of consumed fuel. In the case of transport and installation phases, it is preferable to directly estimate the total value of both indicators derived from the complete programme in the estuary region of Galicia (4080 GAR units):(5)$$Total-GWP_{Transport-Installation}=Total-GWP_{Road}+Total-GWP_{Maritime}$$
(6)$$Total-CED_{Transport-Installation}=Total-CED_{Road}+Total-CED_{Maritime}$$

In Equation (5), *Total − GWP_Transport−Installation_* is the total number of kg of CO_2-eq._ emitted during the transport and installation phases of all GAR units. *Total − GWP_Road_* measures the number of kg of CO_2-eq._ emitted during the ground operations derived from the complete enhancement programme in the estuary region of Galicia (4080 GAR units). *Total − GWP_Maritime_* is equivalent to *Total − GWP_Road_* but, this time, for the maritime operations. Similarly, *Total − CED_Transport-Installation_* is measured in MJ and it is associated with the transport and installation phases of all GAR units (Equation (6)). Consequently, it is the sum of the energy consumed during ground (*Total − CED_Road_*) and maritime operations (*Total − CED_Maritime_*).

Equation (7) is used for estimating *Total − GWP_Road_*:(7)$$Total-GWP_{Road}=\overset{4}{\underset{i=1}{\sum }}\overset{N_{i}}{\underset{j=1}{\sum }}Ld_{i,j}\cdot n_{i,j}\cdot GWP_{km}$$
where *N_i_* is the number of estuaries *j* belonging to zone *i*. It can be deducted from Table 3. As previously indicated *Ld_i,j_* is the distance between the manufacturing plant and the harbour and it is measured in km. Parameter *n_i,j_* is the number of trips needed to transport all GAR units (Table 3) linked to estuary *j* belonging to zone *i*. Finally, *GWP_km_* is the emission factor for 1 km of road transport and it is measured in kg CO_2-eq._/km. Similarly, Equation (8) is used for estimating *Total − GWP_Maritime_*:(8)$$Total-GWP_{Maritime}=\overset{4}{\underset{i=1}{\sum }}\overset{N_{i}}{\underset{j=1}{\sum }}\overset{N_{j}}{\underset{k=1}{\sum }}Md_{i,j,k}\cdot n_{i,j,k}\cdot GWP_{Nm}+T_{i,j,k}\cdot n_{i,j,k}\cdot GWP_{h}$$

In Equation (8) *N_j_* is the number of GAR groups *k* belonging to estuary *j* and it can be deducted from Table 3. *Md_i,j,k_* is the distance between the harbour (*H_i,j_*) and the point of arrival (*PA_i,j,k_*) for a specific GAR group (*A_i,j,k_*). It is measured in nautical miles (Nm). Parameter *n_i,j,k_* is the number of voyages needed to transport all the GAR units (60 units, *A_i,j,k,l_*) belonging to a specific GAR group (*A_i,j,k_*). *GWP_Nm_* is the emission factor for 1 Nm of maritime transport and it is measured in kg CO_2-eq._/Nm. *T_i,j,k_* is the time in hours necessary for loading, unloading and installing the GAR units transported in each voyage and, for calculation purposes, it can be split between the outward and return trips. Finally, *GWP_h_* is the emission factor per hour for the loading, unloading and installation operations and it is measured in kg CO_2-eq._/h. *Total − CED_Road_* can be estimated by using Equation (9):(9)$$Total-CED_{Road}=\overset{4}{\underset{i=1}{\sum }}\overset{N_{i}}{\underset{j=1}{\sum }}Ld_{i,j}\cdot n_{i,j}\cdot CED_{km}$$
where *CED_km_* is the amount of energy consumed per km in road transport, measured in MJ/km. Similarly, *Total − CED_Maritime_* is calculated through Equation (10):(10)$$Total-CED_{Maritime}=\overset{4}{\underset{i=1}{\sum }}\overset{N_{i}}{\underset{j=1}{\sum }}\overset{N_{j}}{\underset{k=1}{\sum }}Md_{i,j,k}\cdot n_{i,j,k}\cdot CED_{Nm}+T_{i,j,k}\cdot n_{i,j,k}\cdot CED_{h}$$

In Equation (10), *CED_Nm_* is the energy use or consumption factor for 1 Nm of maritime transport and it is measured in MJ/Nm. *CED_h_* is the energy factor per hour for the loading, unloading and installation operations and it is measured in MJ/h.

With the objective of applying Equations (7)–(10), it is necessary to know the values that the different emission and energy use factors adopt. *GWP_km_*, in fact, depends on the type of vehicle used for the road transport. As previously indicated in Section 2.2, two different vehicles are considered in this study. The type of fuel consumed by the truck also affects the emissions. Diesel and liquefied natural gas (LNG) are the two options considered in this study. Furthermore, the emissions also vary depending on whether the truck is loaded with GAR units or whether it is a return trip with no cargo, since the consumption of fuel is not the same. In this case, the number of trips with cargo matches the trips without cargo, and the sum of both for a specific estuary is equal to *n_i,j_*. Consequently, it is not necessary to distinguish between trips with and without cargo, since *GWP_km_* can be the average value for the two possible types of trip, as is the case in this study.

Similarly, *GWP_Nm_* also depends on the type of vessel used for the maritime transport (Table 2). The type of fuel consumed by the vessel and if the vessel is empty or full of GAR units are two factors that affect the emissions. Once again, the number of voyages with cargo matches the voyages without GAR units, and the sum of both for a specific GAR group is equal to *n_i,j,k_*. Therefore, *GWP_Nm_* is an average value for the two possible types of voyage. *GWP_h_* also changes with the type of vessel and its fuel. *CED_km_*, *CED_Nm_* and *CED_h_* are analogous to *GWP_km_*, *GWP_Nm_* and *GWP_h_*, respectively, but this time for cumulative energy demand.

Table 7 includes the values for *GWP_km_* and *CED_km_* used in this study. On the other hand, Table 8 contains the values for *GWP_Nm_*, *CED_Nm_*, *GWP_h_*, and *CED_h_*. The reader can find in Carral et al. [29] more information about these factors.

In this study, it is assumed that an articulated truck can transport 3 GAR units in each trip while a rigid truck can only transport 2 GAR units. As previously indicated in Table 2, mini bulk carrier can transport 60 GAR units while the service work boat can only transport 15 GAR units per voyage. Consequently, depending on the estuary *j* (in particular, the number of GAR units included in Table 3) and also depending on the type of vehicle and vessel, the values that *n_i,j_* and *n_i,j,k_* adopt change. Nevertheless, it is possible to estimate them with the information provided so far. On the other hand, mini bulk carrier makes it possible to install (positioning and anchoring) 4 GAR units per hour in their final locations (*PA_i,j,k_*), while the special service work boat allows the installation of 6 GAR units per hour. Furthermore, the loading process in the port, in terms of time consumed, is the same for both types of vessel. In other words, it is possible to load 15 GAR units per hour. This process includes stowage of cargo. Consequently, both types of vessels will need 272 h for such a purpose. Therefore, the values that *T_i,j,k_* adopt for each type of vessel can also be determined. Moreover, in this study it is assumed that the distance between a specific harbour (*H_i,j_*) and the point of arrival (*PA_i,j,k_*) is always the same: 3 Nm. Therefore, *Md_i,j,k_* is always equal to 3 Nm.

With the information provided so far, it is possible to perform certain intermediate calculations that will be used to obtain the results presented in the following sections. By way of example, it is possible to estimate the total number of nautical miles that each type of vessel has to travel for transporting all the GAR units. In the particular case of the mini bulk carrier, 68 outbound trips are needed for transporting all the GAR units (4080/60 = 68). This implies that there will also be 68 return trips, that is, 136 voyages of 3 Nm. Consequently, mini bulk carrier travels 408 Nm (136 × 3 = 408); $\overset{4}{\underset{i=1}{\sum }}\overset{N_{i}}{\underset{j=1}{\sum }}\overset{N_{j}}{\underset{k=1}{\sum }}Md_{i,j,k}\cdot n_{i,j,k}=408$. Similarly, the service work boat travels 1632 nautical miles for the complete restoration of the Galician estuaries. In a similar vein, in the case of the mini bulk carrier 15 h are needed for the installation of the 60 GAR units transported in each voyage. Thus, 1020 h (15 68 = 1020) are consumed during the installation process of the 4080 GAR units. To this must be added the hours required for the loading process in the port (4080/15 = 272 h). Therefore, the total time needed for loading, unloading and installing all GAR units is 1292 h (1020 + 272 = 1292). In other words, $\overset{4}{\underset{i=1}{\sum }}\overset{N_{i}}{\underset{j=1}{\sum }}\overset{N_{j}}{\underset{k=1}{\sum }}T_{i,j,k}\cdot n_{i,j,k}=1292$. Similarly, the second type of vessel needs 952 h for the same processes.

Finally, the total value for *GWP* (kg CO_2-eq._) and *CED* (MJ) derived from the complete restoration of Galician estuaries can be estimated by using the following equations:(11)$$Total-GWP=Total-GWP_{Manufaturing}+\;Total-GWP_{Transport-Installation}$$
(12)$$Total-CED=Total-CED_{Manufaturing}+\;Total-CED_{Transport-Installation}$$

## 3. Results and Discussion

### 3.1. Location of the Manufacturing Plants

Table 9 includes the coordinates of the industrial areas closest to the centres of gravity obtained by the methodology described in Section 2.4 [20]. The information is provided for the two scenarios considered in this study: (i) *MP_i_* and (ii) *CMP*. The coordinates provided for *MP_i_* refer to the following industrial estates: Xove (near Celeiro), Espíritu Santo (south of Lorbé), Cee (outside Corcubión), and Nantes (on the outskirts of Portonovo). The coordinates for *CMP* are the ones of Santa Comba industrial estate. The distances (*Ld_i,j_*) between the manufacturing plant and the corresponding harbours are also shown in Table 9. They are used in Equations (7) and (9).

With the information included in Table 9 together with that provided in Table 3 and in Section 2.5, it is possible to estimate the total distance (round trips) that each type of truck travels for transporting 4080 GAR units: $\overset{4}{\underset{i=1}{\sum }}\overset{N_{i}}{\underset{j=1}{\sum }}Ld_{i,j}\cdot n_{i,j}$. In the case of *MP_i_*, the articulated truck travels 83,600 km while the rigid one travels 125,400 km. On the other hand, for the centralised manufacturing plant (*CMP*), $\overset{4}{\underset{i=1}{\sum }}\overset{N_{i}}{\underset{j=1}{\sum }}Ld_{i,j}\cdot n_{i,j}$ takes a value of 136,160 and 204,240 km for the articulated and rigid options, respectively.

### 3.2. Results for the Manufacturing Process

The results for the manufacturing process are included in Table 10. Data from Table 4, Table 5 and Table 6 as well as Equations (1)–(4) were used for their calculation.

It is also possible to show the results separately for each one of the subprocesses contained in Table 4 as can be seen in Table 11 and Table 12:

### 3.3. Results for the Transport and Installation Phases

The reader can find in Table 13 and Table 14 the results for land transport linked to the complete restoration of the Galician coastal ecosystems, using diesel and LNG as fuel, respectively. A distinction was made among the two type of trucks and the two possible scenarios for the manufacturing plants’ locations (*MP_i_* and *CMP*).

From the results presented in Table 13 and Table 14, it is possible to say that LNG is a better alternative than diesel for both *GWP* and *CED* indicators. This is also true for maritime transport (Table 15). These results are aligned with [32,33,34,35]. For a specific type of fuel and for the same manufacturing model (*MP_i_* or *CMP*), the articulated truck always outperforms the rigid one. It is important to remember that the articulated truck can transport 1 GAR unit more than the rigid option per trip. Consequently, the distance to be covered by the articulated truck is considerably lower than that for the rigid truck. In other words, the difference in the kilometres between the two types of trucks is so large that the rigid option never performs better for a specific scenario, although it presents the lowest emission and energy consumption factors (Table 7). In the same line, the centralised manufacturing plant (*CMP*) is associated with a longer transport distance in comparison with the other manufacturing option (*MP_i_*). Consequently, *GWP* and *CED* results are always better for the *MP_i_* scenario.

Similar results are included in Table 15 for the maritime operations, distinguishing between the two types of vessels and also between the two types of fuel considered in this study.

From the results included in Table 15, once again, it is also possible to say that, for a specific type of vessel, LNG is a better option than diesel in terms of both *GWP* and *CED* indicators [32,33,34,35]. On the other hand, the reader should bear in mind that the figures included in Table 15 take into account the impacts derived from: (i) distance travelled by the vessel (mini bulk carrier: 408 Nm, service boat: 1632 Nm) and, (ii) the time used for port operations and the installation of GAR units (mini bulk carrier: 1292 h, service boat: 952 h). By way of example, the total amount of equivalent CO_2_ emitted by the mini bulk carrier (diesel) is the sum of emissions from transport (128,377.2 kg CO_2-eq._) and the emissions from the other activities (384,860.96 kg CO_2-eq._). In all cases, the time used for port operations and the installation of GAR units presents a larger contribution to *GWP* and *CED* indicators than the maritime transport itself. This is the reason why, for a specific type of fuel, the service work boat always presents better results than mini bulk carrier. This is very relevant since the service work boat has to travel four times the distance covered by the mini bulk carrier, while it only presents a time reduction of about 25% compared to mini bulk carrier. In other words, the time spent on installation and port operations resulted to be more relevant than the maritime distance. While in road travel the higher load capacity stands out, the superior manoeuvrability of the vessel is the deciding factor for maritime transport.

### 3.4. General Results

Multiple scenarios can be defined by combining the different options for the electricity grid mix, the type of truck and its fuel, the type of vessel and its fuel, and the type of manufacturing plant model (one per zone or centralised). Nevertheless, from all the potential scenarios, what is important is to select the combination that produces the least impact in terms of both *GWP* and *CED*. At the same time, it is also interesting to know the value that these two environmental indicators adopt for the worst option. Table 16 includes the information associated with the best and worst possible alternatives for both the Spanish electricity grid mix and the renewable one.

The *GWP* and *CED* results for the scenarios defined in Table 16 are included in Table 17. The reader can also find in Figure 5 the global results per functional unit for the same scenarios.

Based on the results shown in Table 17, it is possible to state that the maritime phase is by far the largest contributor to both *GWP* and *CED* indicators. Although there is some difference between the worst and best scenarios, the improvement cannot be said to be as significant (in relative terms) as in other phases. Manufacture is a relevant stage under the Spanish electricity grid mix scenario. Nevertheless, under the alternative electricity grid mix, manufacturing 4080 GAR units hardly contribute to both impacts compared to the rest of phases analysed in this study. In other words, the use of renewables in the manufacturing process is an obvious way to reduce both *GWP* and *CED*. Nevertheless, the electricity grid mix of a specific country depends on economic, social and geopolitical issues that are not necessarily in line with environmental targets. It is important to note that the results for road transport change considerably from one scenario to another. For the two worst case scenarios, its contribution to *GWP* and *CED* is about 1.8 and 2.8 times (*GWP* and *CED*, respectively) lower than the ones for maritime operations. Despite this, it is by no means negligible. However, for the two best scenarios, road transport contributes around 4.5 and 6 times (*GWP* and *CED*, respectively) less than maritime phases. This highlights the importance of selecting the best scenario for road transport. 

## 4. Conclusions

In this study, the global warming potential (*GWP*) and the cumulative energy demand (*CED*) indicators were analysed for the complete coastal ecosystem enhancement in Galicia (North-western Spain) by using green artificial reefs (GAR units). In particular, manufacture, transport and installation phases were studied. Different alternatives were considered in terms of energy sources, location of the manufacturing plants and transport. Consequently, it was possible to determine the best scenario. The most important conclusions drawn from this study are:The results show that the phase that has the greatest potential impact in contributing to *GWP* and *CED* is the maritime one. Road transport presents a considerable variability between the worst and best possible scenarios.The use of renewables in the electricity grid mix has a positive impact on the manufacturing of GAR units in terms of both *GWP* and *CED* indicators.In terms of road transport, the key factor is to reduce the distance. Consequently, production by zones and the articulated trailer resulted to be the best options (*GWP* and *CED*), since they imply the shortest distance transport.Manoeuvrability resulted to be more important than distance in terms of maritime transport and operations. Consequently, the special work boat obtained better results than mini bulk carrier for *GWP* and *CED* indicators.The use of LNG as fuel for vessel and truck engines lead to reductions in *GHG* emissions and energy consumption.

As for future work, it would be extremely worthwhile to continue exploring ways to optimise maritime transport and to improve the design of auxiliary machines used during the installation of GAR units. Other lines of work can be focused on studying a greater number environmental indicators.

## Figures and Tables

**Figure 1 ijerph-18-01909-f001:**
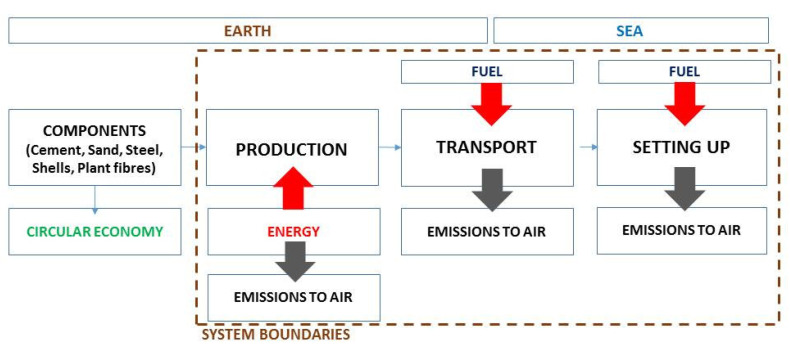
Logistic planning behind creating a GAR, outlining the system’s boundaries, energy needs of the process, and GHG emissions.

**Figure 2 ijerph-18-01909-f002:**
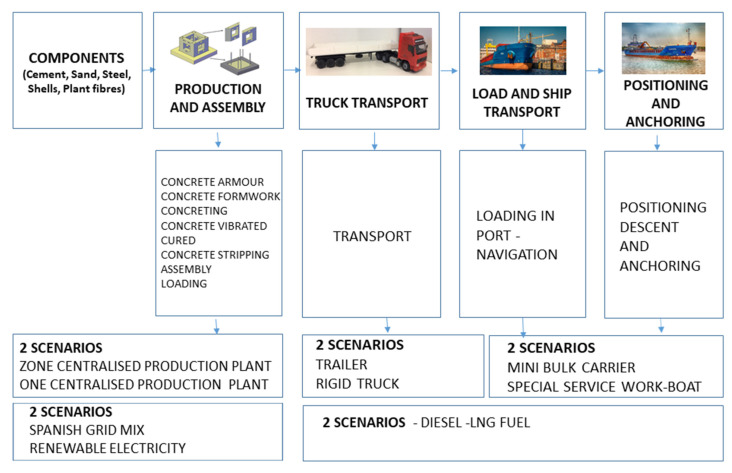
GAR (green artificial reef) units supply chain, including possible scenarios at each stage.

**Figure 3 ijerph-18-01909-f003:**
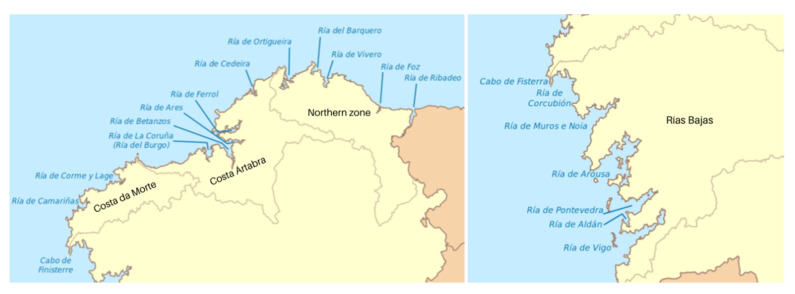
Map of the Galician estuaries.

**Figure 4 ijerph-18-01909-f004:**
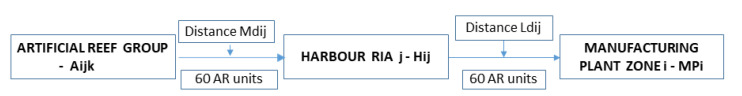
General scheme with the location of different manufacturing plants, one harbour and the point of arrival for a specific green artificial reef (GAR) group.

**Figure 5 ijerph-18-01909-f005:**
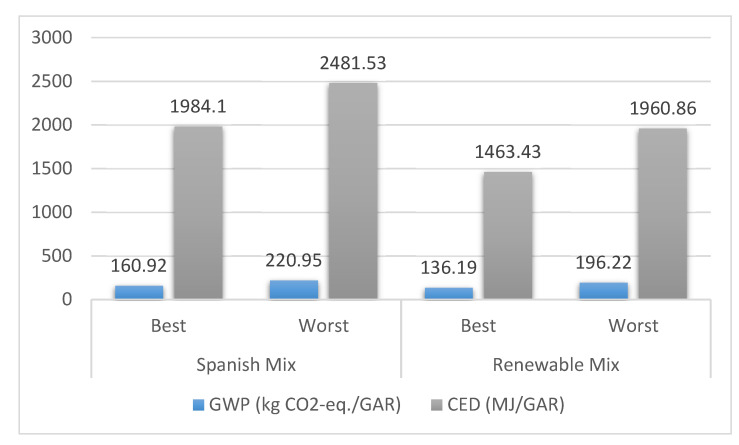
*GWP* (global warming potential) and *CED* (cumulative energy demand) results per functional unit for the scenarios included in Table 17.

**Table 1 ijerph-18-01909-t001:** GAR ^1^ options proposed by authors in [5].

GAR	Conventional Materials	Dosage kg/m^3^	Substitution Materials (waste)	Percentage of Substitution (mass)
Option 1	Cement	350	Oyster and mussel shells	10%
	Sand	750	-	-
	Gravel	1300	-	-
	Frames	45	Eucalyptus vegetable fibres	25%
Option 2	Cement	350	-	-
	Sand	750	Oyster and mussel shells	20%
	Gravel	1300	-	-
	Frames	45	Eucalyptus vegetable fibres	25%
Option 3	Cement	350	Oyster and mussel shells	5%
	Sand	750	Oyster and mussel shells	10%
	Gravel	1300	-	-
	Frames	45	Eucalyptus vegetable fibres	25%
Option 4 ^2^	Cement	350	Oyster and mussel shells	5%
	Sand	750	Oyster and mussel shells	10%
	Gravel	1300	-	-
	Frames	45	Eucalyptus vegetable fibres	100%

^1^ Green artificial reef; ^2^ Mechanical tests are needed before this option can be considered for practical application.

**Table 2 ijerph-18-01909-t002:** Types of vessels deployed for maritime transport.

Type	Dwt ^1^	Vol (m^3^)	Hold/Deck Area (m^2^)	Lpp (m) ^2^	Propulsion Power (Kw) and Speed (Kn)	Cranes Safe Working Load (SWL) Scope	Useful Cargo
Mini bulk carrier	3600	4600	577	81	1470-12Kn	2x12t 18 m	3000 t ^3^
Special service work boat	500	0	180 ^1^	24.2	1310-10Kn	2x12t 12 m	115 t ^4^

^1^ Deadweight tonnage. ^2^ Length between perpendiculars. ^3^ Hold capacity can support 60 GAR units. ^4^ No cargo holds, but with on-deck storage for 15 GAR units.

**Table 3 ijerph-18-01909-t003:** Most relevant information about the Galician estuaries considered in this study.

Zone, *i*	Estuary, *j*	Harbour, *H_i,j_*	GAR Groups, *A_i,j,k_*	Total Number of GAR Units
Northern, *i* = 1	Ribadeo, *j* = 1	Ribadeo, *H_1,1_*	*A_1,1,1_, A_1,1,2_, A_1,1,3_*	180
	Foz, *j* = 2	Foz, *H_1,2_*	*A_1,2,1_, A_1,2,2_, A_1,2,3_*	180
	Vivero, *j* = 3	Celeiro, *H_1,3_*	*A_1,3,1_, A_1,3,2_, A_1,3,3_*	180
	Barquero, *j* = 4	Barquero, *H_1,4_*	*A_1,4,1_, A_1,4,2_, A_1,4,3_*	180
	Ortigueira, *j* = 5	Cariño, *H_1,5_*	*A_1,5,1_, A_1,5,2_, A_1,5,3_*	180
	Cedeira, *j* = 6	Cedeira, *H_1,6_*	*A_1,6,1_, A_1,6,2_, A_1,6,3_*	180
Costa Ártabra, *i* = 2	Ferrol, *j* = 1	Lorbé, *H_2,1_*	*A_2,1,1_, A_2,1,2_, A_2,1,3_, A_2,1,4_*	240
	Ares, *j* = 2	Lorbé, *H_2,2_*	*A_2,2,1_, A_2,2,2_, A_2,2,3_*	180
	Betanzos, *j* = 3	Lorbé, *H_2,3_*	*A_2,3,1_, A_2,3,2_, A_2,3,3_, A_2,3,4_*	240
	La Coruña, *j* = 4	Lorbé, *H_2,4_*	*A_2,4,1_, A_2,4,2_, A_2,4,3_*	180
Costa da Morte, *i* = 3	Corme y Lage, *j* = 1	Corme, *H_3,1_*	*A_3,1,1_, A_3,1,2_, A_3,1,3_*	180
	Camariñas, *j* = 2	Muxía, *H_3,2_*	*A_3,2,1_, A_3,2,2_, A_3,2,3_*	180
	Corcubión, *j* = 3	Corcubión, *H_3,3_*	*A_3,3,1_, A_3,3,2_, A_3,3,3_*	180
	Muros, *j* = 4	Muros, *H_3,4_*	*A_3,4,1_, A_3,4,2_, A_3,4,3_, A_3,4,4_*	240
	Noya, *j* = 5	O Freixo, *H_3,5_*	*A_3,5,1_, A_3,5,2_, A_3,5,3_, A_3,5,4_*	240
Rías Bajas, *i* = 4	Arosa, *j* = 1	Villanueva de Arosa, *H_4,1_*	*A_4,1,1_, A_4,1,2_, A_4,1,3_, A_4,1,4_, A_4,1,5_*	360
	Pontevedra, *j* = 2	Portonovo, *H_4,2_*	*A_4,2,1_, A_4,2,2_, A_4,2,3_, A_4,2,4_, A_4,2,5_*	360
	Aldán, *j* = 3	Cangas, *H_4,3_*	*A_4,3,1_, A_4,3,2_, A_4,3,_*	180
	Vigo, *j* = 4	Cangas, *H_4,4_*	*A_4,4,1_, A_4,4,2_, A_4,4,3_, A_4,4,4_*	240

**Table 4 ijerph-18-01909-t004:** Manufacturing a GAR unit: processes, energy needs and other inputs.

Supply Chain Number, *t*	Processes	Electricity Input (*EI_t_*) (kWh/FU)	Other Inputs	Other Inputs (kg)
1	Concrete armour	2	Iron–AR reinforcement	280.5
2	Concrete formwork ^1^	0.5	Wood	360
3	Concreting	7.5	Aggregates	
4	Transport	4.1	-	-
5	Concrete vibrating	1	-	-
6	Curing	-	-	-
7	Concrete stripping	1	-	-
8	Assembly	50	-	-
9	Loading	16.5	-	-
Total	-	82.6	-	-

^1^ The same formwork can be used for 4 green artificial reef (GAR) units.

**Table 5 ijerph-18-01909-t005:** Spanish electricity grid mix and emission factors for *GWP* and *CED* at the time of producing 1 kWh.

Sources	Share_source_ (%)	Emission Factors (Electricity)	
		*GWP_source_*^6^ (kgCO_2-eq._/kWh)	*CED_source_*^7^ (MJ/kWh)
Natural gas	20.859	0.5 ^1^	7.75 ^3^
Nuclear	20.312	0.005 ^1^	12.024 ^5^
Wind	18.565	0.008 ^1^	0.04 ^3^
Coal	14.363	1.1 ^1^	11.26 ^3^
Hydropower	13.427	0.007 ^1^	0.043 ^3^
Oil	5.276	0.9 ^1^	13.16 ^3^
Solar photovoltaic	2.745	0.07 ^1^	0.65 ^3^
Biofuels	2.004	0.1 ^1^	0.75 ^3^
Solar thermal	1.897	0.203 ^2^	1.15 ^4^
Waste	0.552	0.147 ^1^	1.71 ^3^
Total	100	*GWP_electricity_* = 0.3218	*CED_electricity_* = 6.4478

^1^ Based on [23], ^2^ Based on [28], ^3^ Based on [26], ^4^ Based on [25,27], ^5^ Based on [24]. ^6^ Global warming potential, ^7^ Cumulative energy demand.

**Table 6 ijerph-18-01909-t006:** Alternative or renewable electricity grid mix and emission factors for GWP and CED at the time of producing 1 kWh.

Sources	Share_source_ (%)	Emission Factors (Electricity)	
		*GWP_source_*^6^ (kgCO_2-eq._/kWh)	*CED_source_*^7^ (MJ/kWh)
Wind	50.677	0.008 ^1^	0.04 ^3^
Hydropower	36.652	0.007 ^1^	0.043 ^3^
Solar photovoltaic	7.493	0.07 ^1^	0.65 ^3^
Solar thermal	5.178	0.203 ^2^	1.15 ^4^
Total	100	*GWP_electricity−ideal_* = 0.0224	*CED_electricity−ideal_* = 0.1443

^1^ Based on [23], ^2^ Based on [28], ^3^ Based on [26], ^4^ Based on [25,27]. ^6^ Global warming potential, ^7^ Cumulative energy demand.

**Table 7 ijerph-18-01909-t007:** Emission and energy use factors for road transport. Based on [30].

Type of Vehicle	*GWP_km_*^1^ (kg CO_2-eq._/km)	*CED_km_*^2^ (MJ/km)
Articulated truck (trailer), diesel	1.43	12.46
Rigid truck, diesel	1.37	10.24
Articulated truck (trailer), LNG	1.16	10.09
Rigid truck, LNG	1.11	8.29

^1^ Global warming potential. ^2^ Cumulative energy demand.

**Table 8 ijerph-18-01909-t008:** Emission and energy use factors for maritime transport. Based on [31,32,33,34].

Table 2.	*GWP_Nm_*^1^ (kg CO_2-eq._/Nm)	*CED_Nm_*^2^ (MJ/Nm)	*GWP_h_*^1^ (kg CO_2-eq._/h)	*CED_h_*^2^ (MJ/h)
Mini bulk carrier (diesel)	314.65	3592.68	297.88	3401.26
Special service work boat (diesel)	139.07	1587.85	253.20	2891.10
Mini bulk carrier (LNG)	303.29	3414.48	287.13	3232.88
Special service work boat (LNG)	134.05	1509.09	244.06	2747.70

^1^ Global warming potential. ^2^ Cumulative energy demand.

**Table 9 ijerph-18-01909-t009:** Location of the manufacturing plants and road transport distances *Ld_i,j_* (km).

Zone, *i*	Estuary, *j*	Harbour, *H_i,j_*	Coordinates, *MP_i_*	Distances *Ld_i,j_* (km), *MP_i_*	Coordinates, *CMP*	Distances *Ld_i,j_* (km), *CMP*
Northern, *i* = 1	Ribadeo, *j* = 1	Ribadeo, *H_1,1_*	*MP_1_* −7.536441; 43.688121 (Xove)	*Ld_1,1_* = 53	CMP −8.791827; 43.026543 (Santa Comba)	*Ld_1,1_* = 204
	Foz, *j* = 2	Foz, *H_1,2_*		*Ld_1,2_* = 29		*Ld_1,2_* = 192
	Vivero, *j* = 3	Celeiro, *H_1,3_*		*Ld_1,3_* = 10		*Ld_1,3_* = 188
	Barquero, *j* = 4	Barquero, *H_1,4_*		*Ld_1,4_* = 26		*Ld_1,4_* = 168
	Ortigueira, *j* = 5	Cariño, *H_1,5_*		*Ld_1.5_* = 61		*Ld_1.5_* = 144
	Cedeira, *j* = 6	Cedeira, *H_1,6_*		*Ld_1,6_* = 70		*Ld_1,6_* = 130
Costa Ártabra, *i* = 2	Ferrol, *j* = 1	Lorbé, *H_2,1_*	*MP_2_* −8.29608; 43.305471 (Espíritu Santo)	*Ld_2,1_* = 12		*Ld_2,1_* = 73
	Ares, *j* = 2	Lorbé, *H_2,2_*		*Ld_2,2_* = 12		*Ld_2,2_* = 73
	Betanzos, *j* = 3	Lorbé, *H_2,3_*		*Ld_2,3_* = 12		*Ld_2,3_* = 73
	La Coruña, *j* = 4	Lorbé, *H_2,4_*		*Ld_2,4_* = 12		*Ld_2,4_* = 73
Costa da Morte, *i* = 3	Corme y Lage, *j* = 1	Corme, *H_3,1_*	*MP_3_* −9.191713; 42.976406 (Cee)	*Ld_3,1_* = 52		*Ld_3,1_* = 43
	Camariñas, *j* = 2	Muxía, *H_3,2_*		*Ld_3,2_* = 17		*Ld_3,2_* = 46
	Corcubión, *j* = 3	Corcubión, *H_3,3_*		*Ld_3,3_* = 4		*Ld_3,3_* = 47
	Muros, *j* = 4	Muros, *H_3,4_*		*Ld_3,4_* = 42		*Ld_3,4_* = 43
	Noya, *j* = 5	O Freixo, *H_3,5_*		*Ld_3,5_* = 48		*Ld_3,5_* = 41
Rías Bajas, *i* = 4	Arosa, *j* = 1	Villanueva de Arosa, *H_4,1_*	*MP_4_* −8.8078; 42.429693 (Nantes)	*Ld_4,1_* = 21		*Ld_4,1_* = 85
	Pontevedra, *j* = 2	Portonovo, *H_4,2_*		*Ld_4,2_* = 6		*Ld_4,2_* = 100
	Aldán, *j* = 3	Cangas, *H_4,3_*		*Ld_4,3_* = 62		*Ld_4,3_* = 122
	Vigo, *j* = 4	Cangas, *H_4,4_*		*Ld_4,4_* = 62		*Ld_4,4_* = 122

**Table 10 ijerph-18-01909-t010:** *GWP* (global warming potential) and *CED* (cumulative energy demand) results for the manufacturing process.

Results	Spanish Electricity Grid Mix	Alternative Electricity Grid Mix
*GWP_Manufacturing_* (kg CO_2-eq._/GAR)	26.58	1.85
*CED_Manufacturing_* (MJ/GAR)	532.59	11.92
*Total − GWP_Manufacturing_* (kg CO_2-eq._)	108,446.4	7548
*Total − CED_Manufacturing_* (MJ)	2,172,967.2	48,633.6

**Table 11 ijerph-18-01909-t011:** *GWP* (global warming potential) and *CED* (cumulative energy demand) results for the different manufacturing subprocesses. Spanish electricity grid mix.

Supply Chain Number, *t*	Processes	*GWP_Manufacturing_* (kg CO_2-eq._/GAR)	*CED_Manufacturing_* (MJ/GAR)	*Total − GWP_Manufacturing_* (kg CO_2-eq._)	*Total − CED_Manufacturing_* (MJ)
1	Concrete Armour	0.6436	12.8956	2625.89	52,614.05
2	Concrete Formwork	0.1609	3.2239	656.47	13,153.51
3	Concreting	2.4135	48.3585	9847.08	197,302.68
4	Transport	1.3194	26.4360	5383.15	107,858.88
5	Concrete Vibrating	0.3218	6.4478	1312.94	26,307.02
6	Curing	-	-	-	-
7	Concrete Stripping	0.3218	6.4478	1312.94	26,307.02
8	Assembly	16.09	322.39	65,647.2	1,315,351.2
9	Loading	5.3097	106.39	21,663.58	434,071.2

**Table 12 ijerph-18-01909-t012:** *GWP* (global warming potential) and *CED* (cumulative energy demand) results for the different manufacturing subprocesses. Alternative electricity grid mix.

Supply Chain Number, *t*	Processes	*GWP_Manufacturing_* (kg CO_2-eq._/GAR)	*CED_Manufacturing_* (MJ/GAR)	*Total − GWP_Manufacturing_* (kg CO_2-eq._)	*Total − CED_Manufacturing_* (MJ)
1	Concrete Armour	0.0448	0.2886	182.78	1177.49
2	Concrete Formwork	0.0112	0.0722	45.70	294.58
3	Concreting	0.1680	1.0823	685.44	4415.78
4	Transport	0.0918	0.5916	374.54	2413.73
5	Concrete Vibrating	0.0224	0.1443	91.39	588.74
6	Curing	-	-	-	-
7	Concrete Stripping	0.0224	0.1443	91.39	588.74
8	Assembly	1.12	7.215	4569.6	29,437.2
9	Loading	0.3696	2.3810	1507.97	9714.48

**Table 13 ijerph-18-01909-t013:** *GWP* (global warming potential) and *CED* (cumulative energy demand) results for the different road transport scenarios, using diesel as fuel.

Results	Road Transport Scenarios			
	Articulated Truck and *MP_i_*	Articulated Truck and *CMP*	Rigid Truck and *MP_i_*	Rigid Truck and *CMP*
*Total**− GWP_Road_* (kg CO_2-eq._)	119,548	194,708.8	171,798	279,808.8
*Total**− CED_Road_* (MJ)	1,041,656	1,696,553.6	1,284,096	2,091,417.6

**Table 14 ijerph-18-01909-t014:** *GWP* (global warming potential) and *CED* (cumulative energy demand) results for the different road transport scenarios, using LNG as fuel.

Results	Road Transport Scenarios			
	Articulated Truck and *MP_i_*	Articulated Truck and *CMP*	Rigid Truck and *MP_i_*	Rigid Truck and *CMP*
*Total**− GWP_Road_* (kg CO_2-eq._)	96,976	157,945.6	139,194	226,706.4
*Total**− CED_Road_* (MJ)	843,524	1,373,854.4	1,039,566	1,693,149.6

**Table 15 ijerph-18-01909-t015:** *GWP* (global warming potential) and *CED* (cumulative energy demand) results for the different maritime scenarios.

Results	Maritime Transport Scenarios			
	Mini Bulk Carrier and Diesel	Mini Bulk Carrier and LNG ^1^	Service Work Boat and Diesel	Service Work Boat and LNG ^1^
*Total**− GWP_Maritime_* (kg CO_2-eq._)	513,238.16	494,714.28	468,008.64	451,114.72
*Total**− CED_Maritime_* (MJ)	5,860,241.36	5,569,988.8	5,343,698.4	5,078,645.28

^1^ liquefied natural gas.

**Table 16 ijerph-18-01909-t016:** Definition of the best and worst scenarios for both the Spanish and the renewable electricity grid mixes.

Parameters	Best Scenario	Worst Scenario
Type of truck	Articulated truck	Rigid truck
Truck fuel	LNG ^1^	Diesel
Manufacturing plant model	*MP_i_*	*CMP*
Type of vessel	Service work boat	Mini bulk carrier
Vessel fuel	LNG ^1^	Diesel

^1^ liquefied natural gas.

**Table 17 ijerph-18-01909-t017:** *GWP* (global warming potential) and *CED* (cumulative energy demand) results for the best and worst possible scenarios.

Results	Spanish Electricity Grid Mix		Alternative or Renewable Electricity Grid Mix	
	Best Scenario	Worst Scenario	Best Scenario	Worst Scenario
*Total − GWP_Manufacturing_* (kg CO_2-eq._)	108,446.4	108,446.4	7548	7548
*Total − GWP_Road_* (kg CO_2-eq._)	96,976	279,808.8	96,976	279,808.8
*Total − GWP_Maritime_* (kg CO_2-eq._)	451,114.72	513,238.16	451,114.72	513,238.16
*Total − GWP_Transport-Installation_* (kg CO_2-eq._)	548,090.72	793,046.96	548,090.72	793,046.96
***Total − GWP* (kg CO_2-eq._)**	**656,537.12**	**901,493.36**	**555,638.72**	**800,594.96**
*Total − CED_Manufacturing_* (MJ)	2,172,967.2	2,172,967.2	48,633.6	48,633.6
*Total − CED_Road_* (MJ)	843,524	2,091,417.6	843,524	2,091,417.6
*Total − CED_Maritime_* (MJ)	5,078,645.28	5,860,241.36	5,078,645.28	5,860,241.36
*Total − CED_Transport-Installation_* (MJ)	5,922,169.28	7,951,658.96	5,922,169.28	7,951,658.96
***Total − CED* (MJ)**	**8,095,136.48**	**10,124,626.16**	**5,970,802.88**	**8,000,292.56**

Bold: final results for GWP and CED indicators respectively.

## Data Availability

Not applicable.

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
