# Peer review of "Greenhouse Gas Emissions and Energy Consumption of Coastal Ecosystem Enhancement Programme through Sustainable Artificial Reefs in Galicia"

_ijerph, 2021, doi:10.3390/ijerph18041909_

Round 1

Reviewer 1 Report

The article is very interesting and it only needs small corrections.

The tables should be revised so that the last row does not remain on a different page and if the table has to be divided, the headings need to be on both pages for clarity of reading.

There are words divided in two lines but they do not respect the syllables (lines 30, 100, etc.).

On line 75 authors indicate that the terms AR and GAR are equivalent, however on line 208 it appears that AR represents a group of GAR. Clarify the use of the terms and if they are equivalent use only one throughout the text to clarify the interpretation.

Lines 366 to 370, specify that the term "install" refers to stowage of cargo, so that the time taken by the port crane to load the AR units (15 AR/hour) and the time taken to stow the cargo should be added together. Indicate the total time of each vessel in port during the operation.

Reviewer 2 Report

Paper Review submitted to International Journal of Environmental Research and Public Health, titled “Greenhouse Gas Emission and Energy Consumption of Coastal Ecosystem Recovery Programme through Sustainable Artificial Reef in Galicia,” written by Carral et al.

The authors analysed the global warming potential (GWP) and the cumulative energy demand (CED) indicators for the complete coastal ecosystem recovery in Galicia (North-western Spain) by using green artificial reefs (GAR units), by emphasizing their manufacture, transport, and installation phases.

This paper is original; hence, its original contribution is promising. The written English is logical; hence, the readers should easily follow the contents such as research purpose (question) and its answers. This paper should be published in the journal because of its original contribution and logical approach to tackle the problem. Some minor comments were made by the reviewer as follows.

1. The authors defined “green artificial reef (GAR)” as an artificial reef in which some of the conventional materials are replaced by waster,” by citing some relevant references. If possible, it is better for the readers to capture the overall features of those GARs, not just looking for the relevant references but finding those in the paper. Thus, it is recommended to include a figure or table containing some of the GARs with some characteristics (e.g., size and waste used).

2. The authors used the term “recovery” in the title and almost everywhere in the paper (the author should describe the full name of “PROARR”). The term ‘recovery’ used by the authors may be close to the meaning of ‘restoration.’ Then, the goal to install artificial reefs may be implied by either of the following three ecological terms i.e., rehabilitation, restoration, or enhancement (Svane and Peterson, 2001). ‘Rehabilitation’ implies that some of the ecological features of the pre-disturbed reef ecosystem are replaced (Pratt, 1994). ‘Restoration’, on the other hand, implies that the reef ecosystem is returned to a condition that it would have had if no disturbance had taken place. ‘Enhancement’ is to replace the original ecosystem with a different ecosystem.

As pointed by Svane and Peterson (2001), accomplishing a restoration requires considerable knowledge of both structural (e.g., number of individuals, species, and kind of species) and functional (e.g., energy flow through the system) variables, and such baseline variables for marine reef ecosystem are generally not available. Accordingly, the lack of structural and functional information is troublesome for marine scientists, making it difficult to define the goals of restoration or rehabilitation efforts. Moreover, a goal of direct restoration may not be possible because we do not know the pre-disturbance condition. That is why the alternative term “enhancement” has been used to view establishment of artificial reefs. Enhancement can play a major role preventing habitat loss and increasing endangered habitat. By far the most artificial reefs in the world are established with the aim of enhancement – predominantly with respect to fish populations. In this respect, the term “habitat enhancement structures (HES)” has been used to include artificial reefs, fish aggregation devices (FADs), and living artificial reefs (restoration or enhancement of shellfish, seagrass, coral etc.).

In this respect, it is recommended to consider a right word selection or discuss the meaning of ‘recovery’ if this recommendation does not digress the authors’ logical way or scope describing the paper.

References)

Pratt, J.R., 1994. Artificial habitats and ecosystem restoration: managing for the future. Bulletin of Marine Science 55, 268–275.

Svane, I., Petersen, J.K., 2001. On the problems of epibioses, fouling and artificial reefs, a review. Marine Ecology 22, 3, 169–188. https://doi.org/10.1046/j.1439-0485.2001.01729.x.
